

# Rational synthesis of total damage during cryoprotectant equilibration: modelling and experimental validation of osmomechanical, temperature, and cytotoxic damage in sea urchin (*Paracentrotus lividus*) oocytes

Dominic J. Olver[1], Pablo Heres[2], Estefania Paredes[2] and James D. Benson[1]

[1] Department of Biology, University of Saskatchewan, Saskatoon, Saskatchewan, Canada
[2] Departamento de Ecología y Biología Animal, ECOCOST Lab, Centro de Investigación Mariña, Universidade de Vigo, Vigo, Spain

Corresponding author
James D. Benson,
james.benson@usask.ca

## ABSTRACT

Sea urchins (*e.g.*, *Paracentrotus lividus*) are important for both aquaculture and as model species. Despite their importance, biobanking of urchin oocytes by cryopreservation is currently not possible. Optimized cryoprotectant loading may enable novel vitrification methods and thus successful cryopreservation of oocytes. One method for determining an optimized loading protocol uses membrane characteristics and models of damage, namely osmomechanical damage, temperature damage (*e.g.*, chill injury) and cytotoxicity. Here we present and experimentally evaluate existing and novel models of these damage modalities as a function of time and temperature. In osmomechanical damage experiments, oocytes were exposed for 2 to 30 minutes in hypertonic NaCl or sucrose supplemented seawater or in hypotonic diluted seawater. In temperature damage experiments, oocytes were exposed to 1.7 °C, 10 °C, or 20 °C for 2 to 90 minutes. Cytotoxicity was investigated by exposing oocytes to solutions of $Me_2SO$ for 2 to 30 minutes. We identified a time-dependent osmotic damage model, a temperature-dependent damage model, and a temperature and time-dependent cytotoxicity model. We combined these models to estimate total damage during a cryoprotectant loading protocol and determined the optimal loading protocol for any given goal intracellular cryoprotectant concentration. Given our fitted models, we find sea urchin oocytes can only be loaded to 13% $Me_2SO$ v/v with about 50% survival. This synthesis of multiple damage modalities is the first of its kind and enables a novel approach to modelling cryoprotectant equilibration survival for cells in general.

## INTRODUCTION

Sea urchins are a long-standing model system in developmental biology, embryology, and aquaculture (*Hogan, 1974*; *Hose, 1985*; *Hagen, 1996*; *Agnello & Roccheri, 2010*). Overfishing has led to a global reduction in sea urchin populations with sea urchin peak landings reducing from 120 thousand tonnes in 1995 to about 75 thousand tonnes in 2017 (*Stefánsson et al., 2017*), while climate change may put many sea urchin species into threatened status (*Byrne et al., 2009*; *Hoegh-Guldberg & Bruno, 2010*; *Byrne & Hernández, 2020*). Successful deployment of biobanking techniques will bolster repopulation and conservation projects by enabling indefinite storage and efficient transfer of gametes (*Adams et al., 2003*; *Gasparrini et al., 2007*; *Rezazadeh Valojerdi et al., 2009*; *Zhou et al., 2010*; *Arav, 2014*). However, for sea urchins, cryopreservation methods of preserving female gametes or embryos of sea urchins are very difficult and often unsuccessful with the highest recovery rate of 10% for fertilized eggs (*Asahina & Takahashi, 1978*) or non-existent for oocytes (*Adams et al., 2003*; *Paredes, 2016*). During cryopreservation cells may undergo volumetric change, exposure to high concentrations of cytotoxic solutions, and exposure to cold temperatures that all potentially led to loss of cell viability (*Pegg, 2015*; *Benson, 2015*; *Fahy & Wowk, 2015*). Surprisingly little is known about the mechanisms behind osmomechanical damage, cytotoxicity, and chill injury in the context of cryobiology, and by extension mathematical models to help predict (and therefore avoid) damages are not widely used or well verified. The development of a comprehensive damage model that includes these major sources of damage during cryopreservation will enable a rational approach to optimized cryopreservation protocols along with illuminating mechanisms of damage.

Vitrification is a common and preferred approach for mammalian oocyte cryopreservation in part due to the avoidance of deleterious ice formation (*Arav, 2014*; *Fahy & Wowk, 2015*). However, vitrification requires solutions containing 40% to 50% cryoprotective agents (CPAs). In these solutions that range from 3 to 9 mol/kg of CPAs, cells usually experience high osmotic pressure gradients during the vitrification protocol. The osmotic gradient forces water out of the cell during loading while membrane permeable CPA diffuses into the cell. This results in initial cell volume loss and then subsequent cell volume recovery as CPA and water enters the cell. The two-parameter R (2P) model is a useful model describing the flux of water and CPA across the membrane and is commonly expanded to include temperature dependence (*Benson, 2015*; *Anderson, Benson & Kearsley, 2020*).

During cryopreservation, the cell experiences multiple modalities of damage including osmomechanical (or simply osmotic) related damage (*e.g.*, cell lysis) and cytotoxic damage (solution toxicity). Usually, in the literature, cells are thought to have osmotic tolerance limits, the limit that a cell can shrink or swell to without loss of function from mechanical stress (*Kashuba Benson, Benson & Critser, 2008*; *Blässe et al., 2012*). In addition to this mechanical stress, cytotoxic damage may result from the presence of CPA in the cell: the longer the CPA is inside the cell, the more cytotoxic damage accumulates, ultimately resulting in irreversible damage and cell death (*Lawson, Mukherjee & Sambanis, 2012*;

*Benson, Kearsley & Higgins, 2012*). Classically, an optimized vitrification protocol is one that achieves a goal intracellular CPA concentration that enables vitrification while minimizing cytotoxicity and avoiding osmotic damage (*Davidson, Benson & Higgins, 2014*; *Benson, 2015*).

*Zawlodzka & Takamatsu (2005)* found that osmotic damage is related to not just maximum osmotic challenge but also exposure time. However, time-dependent osmotic damage is poorly understood; while there is no direct evidence for the mechanism of osmotic damage, Zawlodzka and Takamatsu hypothesize there is ultrastructural alteration to the cell as well as reduction of total membrane while the cell is in a reduced volume and attributed this to a membrane regulation hypothesis (*Morris & Homann, 2001*). Such changes to the cell membrane along with alterations to the cytoskeleton may result in catastrophic damage to the cell upon return to isotonic volume. Since these ultrastructural alterations, membrane adaption, and cytoskeletal changes are thought to be time-dependent, it follows that the associated damage is also time-dependent. Indeed, osmotic damage has been found to correlate to the accumulation of volumetric deviance throughout time (*Liu et al., 2009*; *Wang et al., 2011*). Interestingly it has been known for over 100 years that hypertonic exposure in sea urchin oocytes induces parthenogenesis, which may be a mechanism aligning with perceived total cryopreservation protocol damage (*Loeb, 1900*).

Chemical mitigation of osmotic damage may be achieved by alterations to the cytoskeleton and cell membrane (*Fujihira, Kishida & Fukui, 2004*; *Horvath & Seidel, 2006*; *Wang et al., 2016*). For example, cytochalasin-B (which inhibits actin filament polymerization) has been used to aid porcine (*Sus domesticus*) oocytes (*Fujihira, Kishida & Fukui, 2004*) and buffalo (*Bubalus bubalis*) oocytes (*Wang et al., 2016*). It is possible that cytochalasin-B reduces mechanical damage by preventing cytoskeleton reformation during the shrink-swell cycle associated with CPA equilibration. Similarly, cholesterol also aids in the vitrification of cells (*Horvath & Seidel, 2006*), possibly due to increased cell membrane fluidity and the regulation of the connectivity between the cell membrane and the cytoskeleton *via* Phosphatidylinositol 4,5-bisphosphate (PIP$_2$) managed connections (*Sheetz, 2001*; *Taglieri, Delfín & Monasky, 2013*; *Senju et al., 2017*). In fact, cells with added cholesterol can exhibit similar membrane dynamics as cells with denatured cytoskeletons (*Sun et al., 2007*). It is possible that the benefits from the addition of cytochalasin-B or cholesterol are achieved by retarding time-dependent reformation of the cytoskeleton during CPA equilibration. If volumetric damage is proportional to the accumulated change in volume throughout time, then we expect a relationship dependent on time of exposure and osmolality that may be mathematically modelled.

Cytotoxicity of CPAs is a key limitation of cryopreservation and the major limitation for vitrification protocols (*Fahy et al., 2004*). Cytotoxic damage accumulates during both the addition and removal of CPAs and is dependent on CPA concentration, time, and temperature (*Elmoazzen et al., 2007*; *Davidson et al., 2015*; *Traversari et al., 2022*). The damage due to CPA toxicity is complex and poorly understood. Despite the complexity, cytotoxicity can still be modelled (*Lawson, Mukherjee & Sambanis, 2012*; *Benson, Kearsley & Higgins, 2012*; *Davidson, Benson & Higgins, 2014*; *Davidson et al., 2015*). The model presented by *Benson, Kearsley & Higgins (2012)* assumes cytotoxicity is accumulated
throughout time and dependent on CPA concentration in the cell. It is well understood that at lower temperatures cytotoxicity is reduced, however flux of CPA across the membrane is also slower (*Davidson et al., 2015*). Therefore, there is a trade-off between the amount of time required to load/unload a cell with CPA and the cytotoxicity accumulated throughout that time. This trade-off can be used rationally to maximize survival during CPA loading protocols by controlling CPA concentration and environmental temperature (*Davidson, Benson & Higgins, 2014*; *Davidson et al., 2015*). Unfortunately, some cells are sensitive to low temperatures, even in the absence of ice (*MacMillan & Sinclair, 2011*; *Lin et al., 2014*; *Bayley et al., 2018*).

Chill injury (or chilling injury) is injury to cells at sub-physiological temperatures in the absence of ice (*Quinn, 1985*; *Overgaard & Macmillan, 2017*; *Bayley et al., 2018*). Chill injury is documented in multiple species but is not well understood (*Quinn, 1985*; *Lin et al., 2014*; *Overgaard & Macmillan, 2017*; *Bayley et al., 2018*; *Jäkel et al., 2021*). Both historic and current models of chill injury use simple sigmoidal fits relating time, and temperature with survival (*Ohyama & Asahina, 1972*; *Nedvěd, Lavy & Verhoef, 1998*; *Bayley et al., 2018*). Importantly, the time frame for these models is in terms of hours or days—much longer than cryobiological protocols that are usually on the order of tens of minutes for oocytes. Chill injury is hypothesized to be caused by the cell membrane undergoing a phase transition at low temperatures and losing membrane integrity during this phase change (*Quinn, 1985*; *Parkin et al., 1989*). Another possible mechanism of chill injury includes ionic instability during cooling (*Koštál, Vambera & Bastl, 2004*; *Koštál, Yanagimoto & Bastl, 2006*; *MacMillan & Sinclair, 2011*). Despite these hypotheses, there has been no conclusive evidence of a known mechanism of chill injury nor a mathematical model capable of predicting chill injury in the time frame of tens of minutes. To our knowledge, there has been no consideration of chill injury in optimizing vitrification or slow cooling protocols. Despite the lack of a mathematical model, equilibrating cells with CPA at low temperatures may be a good method for avoiding the increased toxicity associated with vitrification protocols (*Davidson, Benson & Higgins, 2014*; *Davidson et al., 2015*). Indeed, such a method is required for an ice-free cryopreservation approach known as liquidus tracking, where the solution concentration is increased to its melting temperature as the system temperature is reduced (*Kay et al., 2015*). However, if too many cells die from chill injury, then this method may not be viable. Therefore, a model that predicts chill injury on the time scale of minutes to tens of minutes is necessary for obtaining an optimized vitrification protocol across both time and temperature.

The object of this article is to propose and experimentally verify novel mathematical models of cell damage during cryopreservation protocols for sea urchin (*Paracentrotus lividus*) oocytes. First, we present and develop multiple damage model types: cytotoxic damage, osmotic damage, and temperature damage (chill injury). Next, we experimentally determine cellular biophysical characteristics including validating the Boyle van't Hoff relation and identifying water and CPA permeability. Using these biophysical models, we experimentally verify our proposed models of osmotic damage, temperature damage, and cytotoxicity. Finally, we combine the damage models into a universal model and

determine an optimized continuous loading protocol as a function of goal intracellular CPA concentrations.

## MATERIALS & METHODS
### Modeling
#### Boyle van't Hoff relation
The Boyle van't Hoff (BvH) relation describes the relationship between equilibrated relative cell volume, $v$, and osmotic pressure, $\Pi$, where $v = V/V_{iso}$, $V$, is cell volume and $V_{iso}$ is cell volume at isotonic pressure, $\Pi_{iso}$ (*van't Hoff, 1887*; *Nobel, 1969*; *Benson, 2012*). Noting the cell is composed of an osmotically active volume and an osmotically inactive volume (solids, large molecules, and bound water), then relative volume may be written in terms of normalized osmotically active fraction, $w_{iso}$, and osmotically inactive fraction, $b$, of the cell (*i.e.*, $v = w_{iso} + b$). The BvH relation takes the form

$$v = w_{iso} \frac{\Pi_{iso}}{\Pi} + b. \tag{1}$$

The fitted, $w_{iso}$, and, $b$, parameters inform other models such as the two-parameter (2P) model to enable time-dependent predictions of volumetric change with respect to osmotic conditions (*Katkov, 2000*; *Benson, Chicone & Critser, 2005*). We note that Eq. (1) does not necessarily need to be normalized with respect to the isotonic point but is done here for simplicity and the direct extraction of $w_{iso}$. See Table 1. For symbols and descriptions.

#### Two-parameter (2P) model
The 2P model describes the flux of water and membrane permeable CPA across the cell membrane at a given moment in time, $t$ (*Jacobs & Stewart, 1932*; *Kleinhans, 1998*). The flux of intracellular water, $W$, may be written in terms of an osmotic pressure gradient, whereas the flux of moles of CPA, $S$, is written in terms of a concentration gradient (*Benson, 2015*). The flux of water and CPA across the cell membrane may be written in terms of a system of ordinary differential equations in the form,

$$\frac{dW}{dt} = -L_p A (\Pi_e - \Pi_i),$$
$$\frac{dS}{dt} = P_s A \left( m_s - \frac{S}{\rho_w W} \right), \tag{2}$$

where $d/dt$ is the derivative with respect to time, $A$ is the surface area of the cell (often assumed constant), $m_s$ is the extracellular molality of membrane permeable CPA, $L_p$ is the hydraulic conductivity, $P_s$ is the CPA permeability constant, and $\rho_w$ is the density of water that we will assume is 1 kg/L. The external osmotic pressure may be written as $\pi_e = RT(\pi_n + \pi_s)$ and the intracellular osmotic pressure may be written as $\pi_i = RT(N_i + S)/(\rho_w W)$, where $R$ is the gas constant, $T$ is temperature, $\pi_n$ is the external osmolality of non-permeable ion/osmolytes, $\pi_s$ is extracellular osmolality of permeable CPA, and $N_i$ is the osmoles of non-permeable internal ions/osmolytes. Typically, the 2P model is fit to experimental data for cell volume throughout CPA equilibration. The
**Table 1  Descriptions of symbols used.**

| Symbol | Description |
| --- | --- |
| $A$ | Surface area of the cell |
| $\alpha$ | Cytotoxicity power constant |
| $\alpha_0$ | Intercept of cytotoxicity power function |
| $\beta$ | Osmotic damage power constant |
| $b$ | Osmotically inactive fraction |
| $C_0$ | Intercept of cytotoxicity proportionality function |
| $C_{osmo}$ | Osmotic damage proportionality constant |
| $C_{temp}$ | Temperature damage proportionality constant |
| $C_{tox}$ | Cytotoxicity proportionality constant |
| $E_\alpha$ | Activation energy (Slope) of cytotoxicity power function |
| $E_{Lp}$ | Activation energy of water |
| $E_{Ps}$ | Activation energy of CPA |
| $E_{tox}$ | Activation energy (Slope) of cytotoxicity proportionality function |
| $J$ | Accumulated damage function |
| $J_{osmo}$ | Osmotic damage function |
| $J_{temp}$ | Temperature damage function |
| $J_{tot}$ | Total damage function |
| $J_{tox}$ | Cytotoxicity damage function |
| $L_0$ | Pe-exponential factor of Arrhenius equation for water |
| $L_p$ | Hydraulic conductivity |
| $m_i$ | Intracellular molality of membrane permeable CPA |
| $m_s$ | Extracellular molality of membrane permeable CPA |
| $N$ | Surviving population |
| $N_i$ | Osmoles of non-permeable internal ions/osmolytes |
| $N_0$ | Initial population |
| $\Pi$ | Osmotic pressure |
| $\Pi_{iso}$ | Isotonic osmotic pressure |
| $\Pi_e$ | Extracellular osmotic pressure |
| $\Pi_i$ | Intracellular osmotic pressure |
| $\pi_n$ | Osmolality of non-permeable ion/osmolytes |
| $\pi_s$ | Extracellular osmolality of permeable CPA |
| $P_0$ | Pre-exponential factor of Arrhenius equation for CPA |
| $P_s$ | CPA permeability constant |
| $\rho_w$ | Density of water |
| $R$ | Gas constant |
| $r$ | Decay rate |
| $S$ | Intracellular moles of CPA |
| $T$ | Temperature |
| $T_{high}$ | High benign threshold temperature |
| $T_{low}$ | Low benign threshold temperature |

**Table 1** (*continued*)

| Symbol | Description |
| --- | --- |
| $T_{phys}$ | Benign threshold temperature |
| $t$ | Time |
| $\tau$ | Time (used for integration up to time $t$) |
| $V$ | Cell volume |
| $V_{iso}$ | Cell volume in isotonic conditions |
| $v$ | Relative cell volume |
| $W$ | Osmotically active volume |
| $w_{iso}$ | Osmotically active fraction |

hydraulic conductivity, $L_p$, and CPA permeability constant, $P_s$, are both assumed to follow an Arrhenius equation, taking the form,

$$
\begin{aligned}
L_p &= L_0 \exp\left[-\frac{E_{lp}}{RT}\right], \\
P_s &= P_0 \exp\left[-\frac{E_{ps}}{RT}\right],
\end{aligned}
\tag{3}
$$

where $L_0$, $P_0$, $E_{lp}$, and $E_{ps}$ are fitted constants. The activation energies $E_{lp}$, and $E_{ps}$ can be determined using linear regression of the log transform of $L_p$ and $P_s$ values as a function of inverse absolute temperature $T^{-1}$.

### Cytotoxicity model

Recent approaches to cytotoxicity models have used a first-order rate equation to model cell death after exposure to $Me_2SO$ (*Elmoazzen et al., 2007*; *Wang et al., 2007*; *Benson, Kearsley & Higgins, 2012*; *Davidson et al., 2015*; *Traversari et al., 2022*):

$$
\frac{dN(t)}{dt} = -rN(t),
\tag{4}
$$

where $N(t)$ is the surviving population size at some time $t$, and $r$ is the decay rate. Solving Eq. (4) with initial population $N_0$ yields

$$
\frac{N(t)}{N_0} = \exp\left[-\int_0^t r(\tau)d\tau\right] = \exp[-J(t)],
$$

where we define $J(t)$ to be the accumulated damage function for the population up to some time $t$ and represented as the integration of the decay rate.

This model may be used to define cytotoxicity, where decay rate is proportional to a power function of intracellular CPA molality $m_i$ (*Benson, Kearsley & Higgins, 2012*; *Traversari et al., 2022*). The relative amount of population surviving at the final time is equal to the damage accrued $J_{tox}$ throughout time such that

$$
J_{tox}(t) = C_{tox} \int_0^t m_i(\tau)^\alpha d\tau,
\tag{5}
$$

where $C_{tox}$ and $\alpha$ are fitted constants, $m_i(t) = S(t)/W(t)$, $S(t)$ and $W(t)$ are intracellular moles of CPA and mass of water at some time $t$ respectively. Davidson and colleagues

(*Davidson et al., 2015*) extend Eq. (5) by allowing $C_{\text{tox}}$ to be a function of temperature governed by the Arrhenius equation (*i.e.*, $C_{\text{tox}}(T) = \exp[C_0 - E_{\text{tox}}/RT]$), taking the form

$$J^1_{\text{tox}}(t, T; C_0, E_{\text{tox}}, \alpha) = \exp\left[C_0 - \frac{E_{\text{tox}}}{RT}\right] \int_0^t \left(\frac{s(\tau)}{w(\tau)}\right)^\alpha d\tau \qquad (6)$$

where $C_0$ and $E_{\text{tox}}$, and $\alpha$ are fitted parameters. The parameter $\alpha$ may be interpreted as a sensitivity parameter, such that highly sensitive cells have a high $\alpha$ value. In practicality, $\alpha$ controls how closely bunched cell survival curves are with respect to intracellular molality as a function of time, while the proportionality parameter $C_{\text{tox}}$ governs the slope of cell survival as a function of time. If cell sensitivity changes with respect to temperature, then Eq. (6) may be expanded by allowing $\alpha$ to be a function of temperature as determined by the Arrhenius equation,

$$J^2_{\text{tox}}(t, T; C_0, E_{\text{tox}}, \alpha_0, E_\alpha) = \exp\left[C_0 - \frac{E_{\text{tox}}}{RT}\right] \int_0^t \left(\frac{s(\tau)}{w(\tau)}\right)^{\exp\left[\alpha_0 - \frac{E_\alpha}{RT}\right]} d\tau, \qquad (7)$$

where $\alpha_0$ and $E_\alpha$ are fitted Arrhenius-type temperature dependence parameters of the toxicity sensitivity parameter $\alpha$.

It is possible the toxicity death rate associated with CPAs is not solely based on intracellular molality of CPAs. Indeed, cryoprotectants such as dimethyl sulfoxide ($Me_2SO$) may create micropores in lipid membranes and change the membrane ion permeability (*Gurtovenko & Anwar, 2007*; *He et al., 2012*; *Fernández & Reigada, 2014*). If damage to the membrane is the key factor for loss of cell function, then the death rate may be better modelled as proportional to the molality of CPA at the cell boundary. Starting from Eq. (7) we can let the decay rate $r$ be proportional to extracellular molality $m_e$ and note that an analytical solution may be found for constant $m_e$:

$$J^3_{\text{tox}}(t, T; C_0, E_{\text{tox}}, \alpha_0, E_\alpha) = \exp\left[C_0 - \frac{E_{\text{tox}}}{RT}\right] m_e^{\exp\left[\alpha_0 - \frac{E_\alpha}{RT}\right]} \times t. \qquad (8)$$

To our knowledge, Eq. (8) is a novel model, however, during unloading or dehydration, the intracellular CPA molality may be more than the extracellular molality and for these cases, Eq. (8) may be modified to take either the maximum or the average of intracellular and extracellular CPA molality.

### Osmotic damage model

Taking a similar approach as above, osmotic damage may be derived using Eq. (4). We propose that the decay rate for osmotic damage is proportional to a power of absolute volumetric change from the isotonic resting state (*i.e.*, $V(t) - V_{\text{iso}}$). We may simplify this relation by normalizing volumetric deviance with respect to isotonic volume; the accumulated osmotic damage function, $J_{\text{osm}}$, takes the form

$$J_{\text{osm}}(t; C_{\text{osm}}, \beta) = C_{\text{osm}} \int_0^t |\Delta v(\tau)|^\beta d\tau, \qquad (9)$$

where $\Delta v(t) = v(t) - 1$, and $C_{\text{osm}}$ and $\beta$ are fitted constants. In this normalized form, $C_{\text{osm}}$ absorbs a $V_{\text{iso}}^\beta$ term and therefore is a different value compared to simple volume

deviance. Note that we employ an absolute difference for change in relative volume since we expect both swelling and shrinking to be harmful. As many cells have different hypotonic compared to hypertonic osmotic sensitivities, we will not assume $C_{osm}$ and $\beta$ are the same for swelling and shrinking (*Kashuba, Benson & Critser, 2014*). Therefore, the constants must be fit independently to hypotonic and hypertonic challenge. Furthermore, it is possible that osmotic damage is also solution-dependent (*e.g.*, high concentrations of sucrose may interact differently with the membrane than NaCl). Finally, note that most existing osmotic tolerance studies assume that osmotic damage is independent of time, whereas our model predicts osmotic damage to be time-dependent.

### Temperature damage model

Similar to the osmotic damage model, we model the temperature damage rate as a function of temperature difference, $\Delta T$, (with reference to physiological temperature, $T_{phys}$), where integrating yields the form:

$$J_{temp}^1 \left(t, T; C_{temp}, T_{phys}\right) = C_{temp} \int_0^t |\Delta T(\tau)| d\tau, \tag{10}$$

and $\Delta T(t) = T(t) - T_{phys}$, $T(t)$ is mean temperature of the cell at time $t$, while $C_{temp}$ and $T_{phys}$ are fitted constants. As in the mechanical damage model (Eq. (9)), note that absolute values are present due to the expectation that cells exposed to above, as well as below, physiological temperatures, will experience damage (and recognizing fitted constants may be different). Furthermore, many organisms may survive normally within a range of temperatures, therefore $T_{phys}$ should be taken as the temperature at which temperature damage starts to occur (for respective cooling or heating). A piecewise function provides a better representation for $\Delta T(t)$ such that

$$\Delta T(t) = \begin{cases} T(t) - T_{low}, & T(t) < T_{low} \\ 0, & T_{low} \leq T(t) \leq T_{high} \\ T(t) - T_{high}, & T_{high} < T(t), \end{cases}$$

where $T_{low}$ and $T_{high}$ are lower and upper bounds of non-damaging physiological temperatures respectively. While the upper bound of physiological temperature is not explored in this article, we do use the lower bound in modelling chill injury and therefore consider only $T_{low}$.

On the time scale of minutes to tens of minutes, it is possible that thermal injury is temperature dependent but not time-dependent (assuming temperature equilibration between cell and environment). For dynamic changes in temperature, the largest change in temperature may be taken. Thus, we take the $L^\infty$ norm of Eq. (10), to get

$$J_{temp}^2 \left(t, T; C_{temp}, T_{phys}\right) = C_{temp} \max[|\Delta T|]. \tag{11}$$

### Total damage model

The proportional survival of a population at the end of a cryopreservation protocol is related to the total damage, $J_{tot}$, of that protocol. Here we assume this to be a linear combination of cytotoxicity, osmotic damage, and temperature damage: $J_{tot} = J_{tox} + J_{osm} + J_{temp}$. This

 

relation is directly derived by taking a first order approximation of population decay in the form

$$\frac{dN(t)}{dt} = -r_{\text{tox}}N(t) - r_{\text{osm}}N(t) - r_{\text{temp}}N(t),$$

where $r_{\text{tox}}$, $r_{\text{osm}}$, $r_{\text{temp}}$, are the decay rates due to toxicity, osmotic damage and temperature, respectively. Integrating provides the expected equality

$$-\log\left[\frac{N(t)}{N_0}\right] = J_{\text{tox}}(t) + J_{\text{osm}}(t) + J_{\text{temp}}(t) = J_{\text{tot}}(t). \tag{12}$$

## Gamete collection

Mature sea urchins (*Paracentrotus lividus*) were obtained from a conditioned broodstock from ECIMAT Marine Science Station at Universidade de Vigo (Spain), maintained for several months in a flow-through system under optimal conditions to enhance gonad maturation. Adults were dissected and gametes were collected directly from the gonads using a Pasteur pipette and transferred into filtered seawater (1 µm and UV sterilized). At least three females per experiment were used to collect oocytes that were subsequently pooled and one male per experiment is used to reduce animal use. Gamete quality was checked before the experiment under light microscope. Oocyte quality was checked with respect to continuous pigmentation and spherical shape (with discontinuous pigmentation and oval oocytes being indicators of poor quality), while sperm quality was checked in terms of motility. Only gamete samples with optimal viewable parameters were selected for experiments.

## Experimental solutions and reagents

Dimethyl sulfoxide ($Me_2SO$), sodium chloride ($NaCl$), and sucrose are from Sigma-Aldrich. Deionized (DI) water was obtained with a Vent Filter MPK01 from Milipore and seawater (SW) was filtered at 1 µm and treated with ultraviolet radiation for sterilization. Where stated, solutions were created by adding the respective solute to seawater taken to be 1.0 osmol/kg. Hypotonic solutions were created by adding deionized water to seawater. Boyle van't Hoff solutions were created by the addition of $NaCl$ with DI water.

## Measurement of viability

Viability (cell survival) was measured using a functional assay (by fertilization and development to the 4-arm pluteus larval stage). Following treatment, sea urchin oocytes were placed in 20 ml food-safe polypropylene vials with room temperature seawater (20 ± 1 °C) and left for approximately 20 minutes to equilibrate before fertilization. Fertilization was done by the addition of sperm (15:1 sperm-oocyte ratio) into containers with oocyte suspension at densities of 20–40 oocytes/ml of seawater (*Saco-Álvarez et al., 2010*; *Paredes, Bellas & Costas, 2015*). After 48 hours, development was arrested by the addition of 40% buffered formalin. Survival was assessed by counting the number of oocytes that develop to the 4-arm pluteus stage (*Saco-Álvarez et al., 2010*; *Paredes, Bellas & Costas, 2015*) over total count (at least 100 per replicate) with a Nikon Eclipse TE2000S microscope with the NIS elements software. The discrimination between normal and abnormal development was

determined under microscope attending to previous work focused on larval development and morphology in our lab and guidelines from other experts in abnormalities in the sea urchin *P. lividus* in ecotoxicological larval bioassays (*Paredes & Bellas, 2009*; *Saco-Álvarez et al., 2010*; *Bellas & Paredes, 2011*; *Carballeira et al., 2012*). Typical larval abnormalities found ranged from: delayed development (larvae without developed arms, gastrulas or blastulas), deviations from the 4-arm pluteus larvae shape like deformations (protruding spines, bent arms, missing arms) or presence of non developed eggs. For each experiment, survival was normalized with respect to the respective control using the same three females and 1 male as selected for the treatments.

## Boyle van't Hoff (BvH) experimental design

NaCl solutions (486, 602, 778, 972, 1215, 1944, 3888 mOsm/kg) were prepared in Milli-Q water. Sea urchin oocytes were concentrated using a 40 $\mu$m mesh and deposited in Petri dishes containing NaCl solutions. Temperature was maintained at $18 \pm 1$ °C throughout all the BvH treatments. Mature oocytes from three females were pooled together and used for experiments on the respective days. Replicates are defined as separate samples from the pooled oocyte population separated into distinct containers and exposed to the same treatment. Three replicates were assayed for each treatment. Oocyte images were captured using an SMZ 1500 binocular loupe and NIS Elements D software at 30 min exposure and diameters of oocytes were obtained. Digital images were analyzed and the lengths of the major and minor semi-axes($d_1$ and $d_2$ respectively) of each oocyte were measured. Each cell volume, $V$, was determined using the formula

$$V = \left(\frac{4}{3}\pi\right)\left(\frac{d_1+d_2}{2}\right)^3,$$

when $\max\{|d_1-(d_1+d_2)/2|, |d_2-(d_1+d_2)/2|\} < 0.1$. If this condition was not met, then the cell volume was calculated assuming prolate spheroid shape and using the following formula:

$$V = \left(\frac{4}{3}\pi\right)\left(\frac{d_1}{2}\right)^2\left(\frac{d_2}{2}\right).$$

## Membrane characteristics experimental design

Sea urchin oocytes were abruptly exposed to seawater with 1.5 mol/kg Me$_2$SO at room temperature ($20 \pm 1$ °C, with daily room temperatures logged explicitly and used in the analysis). Oocyte volume was recorded with a Nikon Eclipse TE2000S microscope for 22 minutes. A time-lapse images series was recorded for at least 51 oocytes. Images were processed in ImageJ (*Schindelin et al., 2012*) to determine the cross-sectional area that was subsequently converted to volumes of individual oocytes at each time step. The 2P model (Eq. (2)) was fit to individual cell volumes throughout time. The Me$_2$SO equilibration protocol was repeated at $10 \pm 1$ °C and $6 \pm 1$ °C and individual oocyte volumes were tracked throughout time. Reduced temperature experiments were performed in an air-conditioned room. $L_p$ and $P_s$ values were fit to the Arrhenius equation *via* a log transformation and linear regression against inverse temperature.

### Osmotic damage experimental design

To test the hypothesis of time-independent osmotic tolerance limits, oocytes were exposed to hypertonic solutions of seawater (UV filtered seawater at 1.0 osmol/kg) supplemented with 0.5, 1.0, and 1.5 osmol/kg of NaCl for duration periods of 2, 6, 15, and 30 minutes. Technician-corrected exposure times were recorded for each treatment(*e.g.*, 2.5 minutes instead of 2 minutes). To determine if hypertonic related damage is solution specific, the experiment was repeated with additions of sucrose in place of NaCl. Sucrose treatments included seawater (1.0 osmol/kg) supplemented with 0.5, 1.0, 1.5, and 2.0 osmol/kg sucrose for the same exposure times. Hypotonic damage was tested similarly, using distilled water diluted seawater at osmolalities of 0.8, 0.7, 0.6, and 0.5 osmol/kg for the same exposure periods. All experiments were done at room temperature (20 $\pm$ 1 °C). Replicates are defined as separate samples from a single homogenized oocyte population in distinct containers exposed to the same treatment. Each treatment had three replicates each containing more than 600 oocytes per replicate (of which at least 100 were counted). Containers consisted of 20 ml food-safe polypropylene vials and solutions were filtered with a 20 $\mu$m polyethylene mesh. The control included 3 replicates (oocytes exposed to seawater). Survival was calculated as described in Measurement of Viability section.

### Temperature damage experimental design

We tested the effects of chill injury at three temperatures: 20 $\pm$ 1 °C and 10 $\pm$ 1 °C in air-conditioned rooms and in an ice bath with temperature 1.7 $\pm$ 1.5 °C (with observed temperature ranges from 0.4 to 3.0 °C). Exposure times were 2, 6, 15, 30, 50, 75, and 90 minutes with three replicates (as described above). All treatments were conducted in seawater (1.0 osmol/kg). Survival was normalized with respect to the mean survival of oocytes throughout the 20 °C treatment. Survival was calculated as described in Measurement of Viability section.

### Cytotoxicity experimental design

We tested the effects of cytotoxicity at three temperatures concurrently with the temperature damage experiment (see above). All treatments were conducted in seawater (1.0 osmol/kg) supplemented with 0.5 osmol/kg, 1.0 osmol/kg, or 1.5 osmol/kg $Me_2SO$. Exposure times were 2, 6, 15, and 30 minutes with three replicates (as described above). Note, for ice bath treatment, only additions of 0.5 osmol/kg, and 1.0 osmol/kg $Me_2SO$ were performed. The $J_{osm}$ model (Eq. (9)) was used to account for any osmotic damage during both loading and unloading of $Me_2SO$. For each respective treatment, survival was normalized to the mean survival of oocytes at the respective temperature in isotonic seawater (1.0 osmol/kg). The cytotoxicity models and the osmotic damage model were informed by the 2P model (Eq. (2)). Survival was calculated as described in Measurement of Viability section.

### Numerical optimization design

We used $J_{tot}$ (Eq. (12)) with parameters from our above experiments to obtain predicted relative population survival for a given CPA equilibration protocol. We initially attempted to determine optimal protocols using a traditional stepwise approach, where we numerically optimized piecewise constant concentration and temperature combinations. We did not

find a combination of loading times, concentrations, and temperatures that resulted in any predicted cell survival at goal concentrations beyond 15 percent CPA. Therefore, drawing from literature suggesting that concentration gradient approaches may be more successful, we hypothesized that a "constant volume" approach similar to that suggested by *Levin (1982)* and extended to time and toxicity optimal controls by Benson and colleagues (*Benson, 2015*; *Davidson et al., 2015*; *Benson et al., 2018*), will provide an optimal loading protocol. Briefly, in these protocols cell volumes are manipulated by continuous control of extracellular permeating and nonpermeating solute concentrations and maintained at a constant value until a goal intracellular concentration is reached. Here we hypothesize that the optimal equilibration strategy is of this class of constant volume *vs* time CPA loading functions and optimize over two control variables: equilibration temperature (considered fixed for the protocol) and loading volume. For cells equilibrated at below isotonic volume, we consider the loading protocol complete when intracellular CPA/water volume ratio (v/v) reaches the goal. For cells equilibrated at above isotonic volume, the cells are equilibrated until a final-step dehydration of the cell back to isotonic volume results in the desired CPA v/v (*Benson, Chicone & Critser, 2012*). This approach thus includes the previously published minimal time, minimal volume change, and minimal cytotoxicity protocols and all protocols with target volumes in between (*Benson, 2015*; *Davidson et al., 2015*; *Benson et al., 2018*). Using the obtained loading protocol concentration *versus* time function, the associated total survival for the respective loading protocol can be obtained with the $J_{tot}$ (Eq. (12)) model (*i.e.*, $\exp[-J_{tot}]$). This defines the two-dimensional optimization problem: for a given target intracellular goal CPA volume fraction, $f$, find

$$\max_{T, V_{target}} \exp[-J_{tot}].$$

Loading target volume, $V_{target}$, was limited to the range of 0.5 to 3, and optima were obtained within this region with a survival of close to zero at the boundaries, indicating the global optima is within these bounds. The temperature is confined to physiological temperature (293 K) down to 273 K. All optima were >283 K indicating global optima are above 273 K. The goal CPA v/v investigated was varied from 0.01 to 0.5 and Mathematica's(12.3v) *NMaximize* function was used to obtain each respective optimum (*Wolfram Research Inc, 2018*).

## Analysis

All models, fits, and analyses were conducted in Mathematica (12.3v) (*Wolfram Research Inc, 2018*). The BvH relation (Eq. (1)) was fit through linear regression using Mathematica's *LinearModelFit* function, and a Durbin Watson score was obtained for each regression (first-order autocorrelation test of residuals) (*Durbin & Watson, 1950*; *Ali, 1987*). A paired T-test between NaCl and sucrose treatments for the osmotic damage experiments was calculated using Mathematica's *PairedTTest* function with normality tested by Shapiro-Wilk test and supplemented with a nonparametric Wilcoxon signed ranked test (which does not assume normality) *via* Mathematica's *SignedRankedTest*. Pairwise comparisons were between grouped mean values of the replicates for the respective treatment (time and osmolality) and omitted if either treatment was missing. Spearman Ranked correlation

tests were performed for each chill damage treatment between temperature and time. All nonlinear models were fit to each respective dataset using the *NonLinearModelFit* function in Mathematica. A grid search was used to provide initial guesses for fitting parameters of Eqs. (6)–(9). Comparison between models is conducted with goodness of fit with respect to the plotted data and respective model along with AIC and BIC. Adjusted $R^2$ is presented for nonlinear models but conclusions must not be taken with respect to Adjusted $R^2$ values alone (*Spiess & Neumeyer, 2010*). We note that we present $R^2$ value for the Arrhenius equation, which, while linear, is a transformed dataset and therefore the presented $R^2$ value must be viewed with this consideration.

Replicates that were erroneously conducted were documented and removed from the analysis. These discarded replicates included some from the NaCl treatments: 1.5 osmol/kg at 30 minutes (one removed out of three replicates); sucrose treatment: 3 osmol/kg for 2 minutes (three out of three replicates; not filtered until ~5 minutes); the Me2SO treatments: 0.5 osmol/kg 10 °C for 2 mins (three out of three replicates), 0.5 osmol/kg at 10 °C for 6 mins (two out of three replicates), 1 osmol/kg in an ice bath for 2 mins (one out of three replicates); the chilling treatment: seawater in an ice bath for 50 mins (one out of three replicates). All other treatments consisted of three replicates. Code and data are available in the supplemental materials.

## RESULTS

### Cell and membrane osmotic characteristics

First, equilibrated volumes of sea urchin oocytes were fit by the Boyle van't Hoff relation (Fig. 1A) with an $R^2$ value of 0.704 ($n = 1103$). The linear relation did not appear autocorrelated (Durbin Watson score of 2.32), indicating Eq. (1) is an appropriate model. The fitted osmotically active fraction, $w_{iso}$, of *P. lividus* oocytes is $0.687 \pm 0.014$ (standard error; SE), and the osmotically inactive fraction, $b$, is $0.311 \pm 0.017$. Next, we recorded exposed oocytes to 1.5 osmol/kg of $Me_2SO$ at 6 °C, 10 °C and 20 °C. We tracked individual oocytes throughout time, and we fit the 2P model to identify initial cell volume, $L_p$, and $P_s$ values ($n = 214$). Best fit $L_p$ were $0.118 \pm 0.005$, $0.044 \pm 0.001$, and $0.041 \pm 0.001$ $\mu m$ $min^{-1}$ $atm^{-1}$ at 20 °C, 10 °C, and 6 °C, respectively. Best fit $P_s$ were $5.036 \pm 0.198$, $0.541 \pm 0.030$, and $0.710 \pm 0.041$ $\mu m$ $min^{-1}$ at 20 °C, 10 °C, and 6 °C, respectively. Figs. 1B–1C shows the results of these fits along with the fitted Arrhenius equation of $\log L_p$ and $\log P_s$ as a function of inverse temperature. The intercepts and slopes (activation energy) of these regressions were $L_0 = \exp[16.1 \pm 1.1]$ $\mu m$ $atm^{-1}min^{-1}$ and $E_{Lp} = 13.0 \pm 0.6$ mol $kcal^{-1}$ for $\log L_p$ and $P_0 = \exp[41.8 \pm 2.0]\mu m$ $min^{-1}$ and $E_{Ps} = 25.9 \pm 1.1$ mol $kcal^{-1}$ for $\log P_s$, with $R^2$ values of 0.97 and 0.98 for each respective linear fit ($n = 214$). Fig. 1C shows the mean of each normalized cell with respect to its initial volume and plotted is Eq. (2) using the average $L_p$ and $P_s$ values for the given temperature. While the 2P model fits well to the data, unexpectedly, oocytes exposed to $Me_2SO$ at 6 °C were larger on average than oocytes exposed at 10 °C.

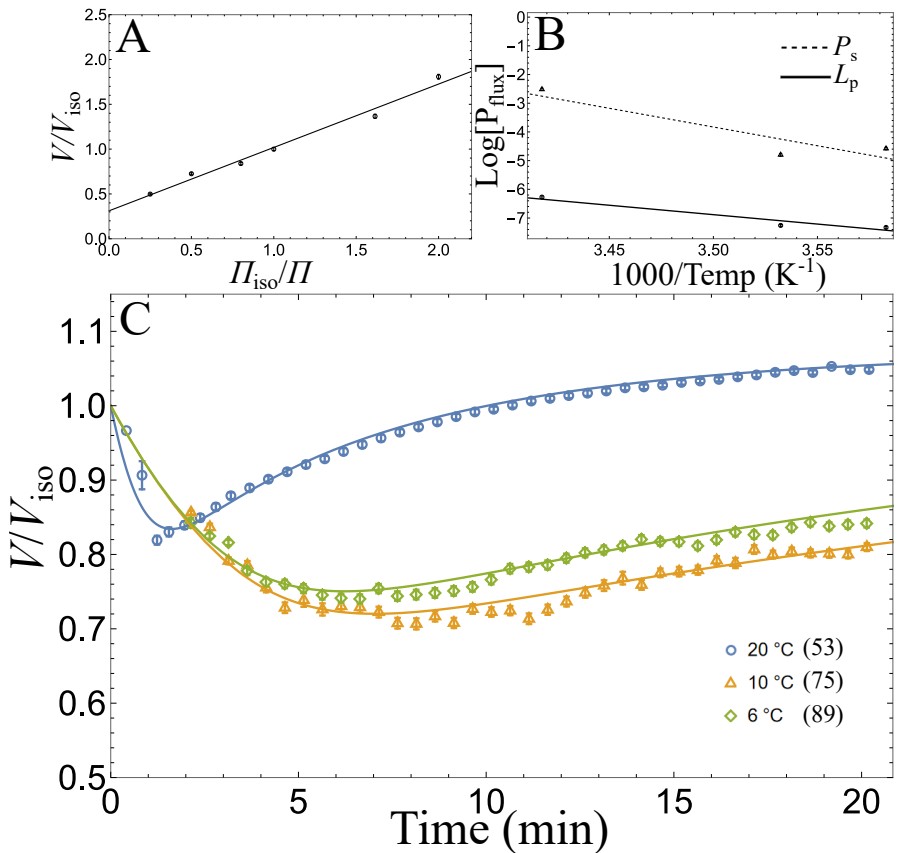

**Figure 1** **Osmotic characteristics of *P. lividus* oocytes.** Symbols represent mean values and error bars represent SEM. Volume, $V$, is normalized with respect to isotonic volume, $V$iso. (A) Nondimensionalized Boyle van't Hoff plot ($n = 1103$). (B) Arrhenius log plot where $P_{flux}$ represents $L_p$ and $P_s$ ($n = 214$). (C) Normalized volume flux of oocytes in seawater (1.0 osmol/kg) with addition of 1.5 osmol/kg dimethyl sulfoxide at differing temperatures. Solid lines are the mean of the individually fitted 2P model with respect to each oocyte ($n = 214$), and the number of oocytes measured at each temperature is given in parentheses.

## Osmotic damage experiments

Using an $L_p$ value in pure seawater of 0.191 µm atm$^{-1}$ (*Adams et al., 2003*), $J_{osm}(t; C_{osm},$ $\beta)$ (Eq. (9)) was fit to seawater (1.00 osmol/kg) supplemented with NaCl (Fig. 2A), with sucrose (Fig. 2B), or diluted with DI water (Fig. 2C). The fitted model appears to fit well in all cases with high adjusted $R^2$ values (adjR$^2$>0.98). Noticeably, shrinking damages for NaCl and sucrose had high power parameters, $\beta = 10.88 \pm 0.54$, and $\beta = 11.29 \pm 0.61$ respectively, while the swelling damage assessed in diluted seawater had a lower fitted power $\beta = 1.55 \pm 0.17$ (See Table 2). Furthermore, the parameters assessed with the NaCl treatment were significantly different from that assessed with the sucrose treatment (parameter confidence region ellipsoids of Eq. (9) do not overlap with a confidence level of 95%). Sucrose treatments tended to have lower survivability when compared to NaCl treatments of the same time and concentration (Figs. 2A–2B). A paired T-test between
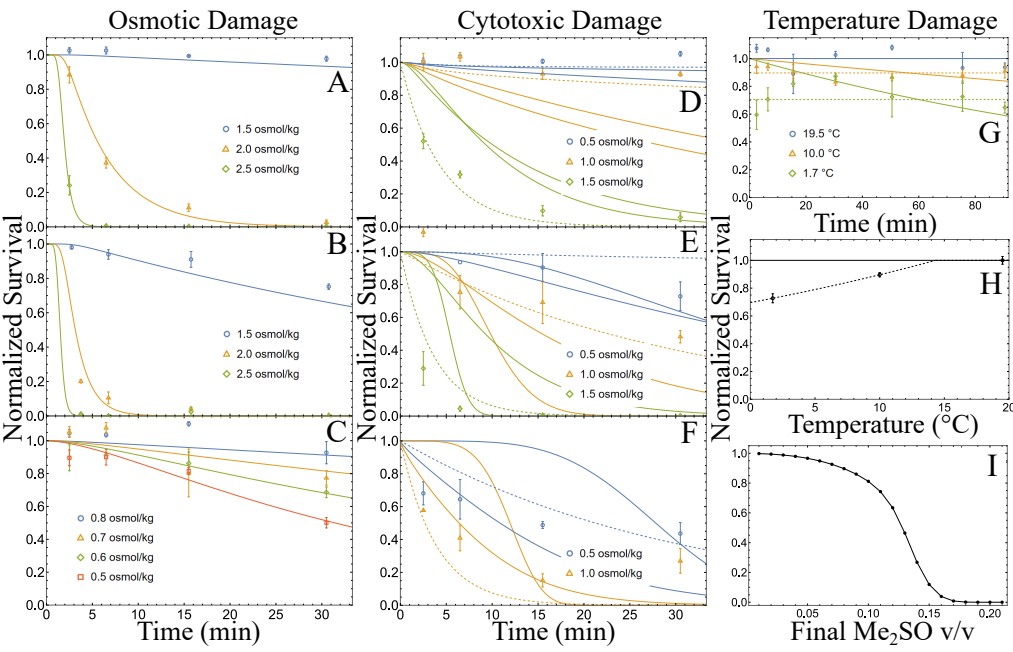

**Figure 2   Damage experiments, model fits, and optimized loading.** (A,B,C) Normalized survival of oocytes in (A) NaCl + seawater ($n = 35$), (B) sucrose + seawater ($n = 48$), and (C) DI water + seawater ($n = 48$). Survival is normalized to the respective controls (1.0 osmol/kg seawater). Symbols represent mean values and error bars represent standard error of the mean (SEM). Solid lines represent the best fit $J_{osm}(t; C_{osm}, \beta)$ model. Note 3.0 osmol/kg sucrose + seawater data are not plotted but all have 0% survival. (D,E,F) Normalized survival of oocytes with respect to time at (D) room temperature (20 ± 1 °C), (E) 10 °C, and (F) 1.7 ± 1.5 °C. Data are normalized to their respective control at their respective temperature ($n = 89$). Listed osmolalities are the added osmolality of dimethyl sulfoxide to isotonic seawater (1.0 osmol/kg). The solid thick line represents $J_{tox}^1(t, T; C_0, E_{tox}, \alpha)$ (Eq. (6)), the solid thin line represents $J_{tox}^2(t, T; C_0, E_{tox}, \alpha_0, E_\alpha)$ (Eq. (7)), the dotted line represents $J_{tox}^3(t, T; C_0, E_{tox}, \alpha_0, E_\alpha)$ (Eq. (8)). (G,H) (G) Normalized survival with respect to time ($n= 62$). Solid lines represent the time and temperature dependent model ($J_{temp}^1(t, T; C_{temp}, T_{phys})$; Eq. (10)), while dotted lines represent the temperature dependent model ($J_{temp}^2(t, T; C_{temp}, T_{phys})$; Eq. (11)). (H) Normalized survival with respect to temperature ($n = 62$). The dotted line represents the temperature dependent model $J_{temp}^2(t, T; C_{temp}, T_{phys})$ (Eq. (11)). Data are normalized with respect to room temperature (measured at 19.5 °C) in 1.0 osmol/kg seawater. (I) Results of a numerical optimal search function for predicted maximal ratio (black dots) of initial to surviving population at the end of an optimal loading protocol with concentrations of Me₂SO. Max survival ratios beyond 0.21 v/v CPA v/v are $< 10^{-7}$.

NaCl (mean = 0.454, SD = 0.487) and sucrose (mean = 0.375, SD = 0.452) treatments showed a significant difference in survival ($t(19) = 2.594$, $p = 0.029$). As the data failed the Shapiro-Wilks test for normality, the Wilcoxon signed ranked test was used and agreed that NaCl (median = 0.241, median deviation = 0.239) and Sucrose (median = 0.073, median deviation = 0.073) significantly differed in survival ($Z = 51.0$, $p = 0.019$). Thus, for a given time and concentration, osmotic damage to an oocyte in a sucrose solution is more damaging than NaCl solution (Figs. 2A–2B).

**Table 2  Comparison of damage models, parameters, and fit quality metrics.**

| Time-Dependent Damage Model | $C_{osmo} \pm SE$ * | $\beta \pm SE$ * | $n$ | adjR$^2$ |
|---|---|---|---|---|
| $J_{osm}(t;C_{osm},\beta)$ (NaCl + SW) | $6.55 \pm 2.55$ | $10.9 \pm 0.5$ | 35 | 0.995 |
| $J_{osm}(t;C_{osm},\beta)$ (Sucrose + SW) | $41.3 \pm 20.6$ | $11.3 \pm 0.6$ | 48 | 0.993 |
| $J_{osm}(t;C_{osm},\beta)$ (DI + SW) | $4.71 \pm 0.90 \times 10^{-4}$ | $1.55 \pm 0.2$ | 48 | 0.986 |

| Cytotoxicity Model | $C_0$ * | $E_{tox}$ (kcal/mol) | $\alpha$ * | $E_\alpha$ (kcal/mol) | $n$ | adjR$^2$ | AIC | BIC |
|---|---|---|---|---|---|---|---|---|
| $J_{tox}^1(t,T;C_0,E_{tox},\alpha)$ (Eq. 6) | $-128.8 \pm 19.2$ | $-70.2 \pm 11.0$ | $4.7 \pm 0.5$ | $n/a$ | 89 | 0.813 | 44.48 | 54.44 |
| $J_{tox}^2(t,T;C_0,E_{tox},\alpha_0,E_\alpha)$ (Eq. 7) | $-44.88 \pm 11.06$ | $-21.55 \pm 6.34$ | $36.97 \pm 8.82$ | $19.75 \pm 4.82$ | 89 | 0.848 | 27.18 | 39.62 |
| $J_{tox}^3(t,T;C_0,E_{tox},\alpha_0,E_\alpha)$ (Eq. 8) | $-80.55 \pm 8.62$ | $-41.01 \pm 4.77$ | $22.20 \pm 1.82$ | $11.53 \pm 1.02$ | 89 | 0.953 | $-78.12$ | $-65.68$ |

| Temperature Damage Model | $C_{temp} \pm SE$ (s/°C) | $T_{low} \pm SE$ (°C) | $n$ | adjR$^2$ | AIC | BIC |
|---|---|---|---|---|---|---|
| $J_{temp}^1(t,T;C_{temp},T_{phys})$ (Eq. 10) | $0.068 \pm 0.026$ | $14.8 \pm 3.3$ | 62 | 0.970 | $-52.47$ | $-46.09$ |
| $J_{temp}^2(t,T;C_{temp},T_{phys})$ (Eq. 11) | $0.025 \pm 0.006$ | $14.3 \pm 1.9$ | 62 | 0.983 | $-87.18$ | $-80.80$ |

**Notes.**
*Dimensionless
DI, Deionized water; SE, Standard Error; SW, Seawater.

## Cytotoxicity damage experiments

We corrected the cytotoxicity survival data by using the fitted osmotic damage model, $J_{osm}(t;C_{osm},\beta)$ (Eq. (9)), to correct for any osmotic damage and using the results from the chill injury experiments, normalized with respect to mean survival at each temperature to account for chill injury. The three cytotoxicity models (Eqs. (6)-(8)) were fit and plotted across temperature and concentration (Figs. 2D–2F). The $J_{tox}^3(t,T;C_0,E_{tox},\alpha_0,E_\alpha)$ model, with decay rate proportional to a power of extracellular CPA molality, performed the best with the highest adjR$^2$ and lowest AIC, and BIC values when compared to $J_{tox}^1(t,T;C_0,E_{tox},\alpha)$ and $J_{tox}^2(t,T;C_0,E_{tox},\alpha_0,E_\alpha)$ models (see Table 2 for respective values and fitted parameters). Furthermore, $J_{tox}^3$ appears to best visually fit data across each temperature (Figs. 2D–2F). Interestingly, the fitted power function decreases for the $J_{tox}^3$ model as temperature decreases, such that $\exp[\alpha_0 - E_\alpha/(RT)] = 11.08$ at 20 °C, 5.51 at 10 °C, and 2.96 at 1.7 °C, while the proportionality function, $\exp[C_0 - E_{tox}/(RT)]$, increases as temperature decreases (see Table 2 for fitted parameters). A similar trend is found for $J_{tox}^2$ (*i.e.*, power function decreases but proportionality function increases).

## Temperature damage experiments

Temperature damage related to chill injury was investigated using the time-dependent and independent versions of $J_{temp}^1(t,T;C_{temp},T_{phys})$ and $J_{temp}^2(t,T;C_{temp},T_{phys})$ (Eqs. (10) and (11)), respectively. Their fitted models are shown in Fig. 2G with respect to time, and just

temperature in Fig. 2F. Performing a Spearman Rank correlation test for 10 °C ($r$ (19) = −0.232, $p$ = 0.316) and at 1.7 °C ($r$ (18) = −0.034, $p$ = 0.888) results in no significant correlation between time and survival. For completeness we performed a similar test at room temperature and as expected did not find any significant correlation between survival and time ($r$ (19) = −0.401, $p$ = 0.071). While both models appear to have high adjusted $R^2$ values (adj$R^2$ ≥0.97), the time-independent model ($J^2_{temp}$; Eq. (11)) outperforms the time-dependent model ($J^1_{temp}$; Eq. (10)) in terms of adj$R^2$, AIC, and BIC (Table 2).

## Total damage model

Using osmotic damage defined by $J_{osm}(t;C_{osm},\beta)$ (Eq. (9)), temperature damage defined by $J^2_{temp}(t,T;C_{temp},T_{phys})$ (Eq. (11)), and cytotoxicity damage defined by $J^3_{tox}(t,T;C_0,E_{tox},\alpha_0,E_\alpha)$ (Eq. (8)), we used $J_{tot}$ (Eq. (12)) to obtain maximal survival during loading. In Fig. 2I we show the survival ratio at the end of the optimal loading protocol with respect to the goal CPA ($Me_2SO$) v/v (values above 0.21 not shown). Sea urchin oocytes have an approximately 50% survival when loading to about 0.13 $Me_2SO$ v/v, compared to having a near zero survival when loading to 0.21 $Me_2SO$ v/v. Vitrification level solutions such as 0.4 and 0.5 $Me_2SO$ v/v, correspond to maximal survivals of 0%.

## DISCUSSION

### Cell and membrane osmotic characteristics

Volume appears to be linearly related to inverse osmotic pressure as predicted for ideal osmometers following the BvH relation (Fig. 1A). This supports the assumption that *Paracentrotus lividus* oocytes are ideal osmometers. Notably, the sea urchin *Evechinus chloroticus* oocyte were also found to be an ideal osmometer (*Adams et al., 2003*).

We recorded many oocytes during equilibration with $Me_2SO$ ($n = 214$) and normalized these oocytes with respect to each individual fitted initial volume. This large sample size with individual oocyte measured $L_p$, $P_s$, and $V_{iso}$ parameters provides a rich dataset with fitted permeability parameters comparable to *E. chloroticus* (*Adams et al., 2003*). Unexpectedly, the mean population volume for *P. lividus* at 10 °C recovered slower than the mean population volume at 6 °C (Fig. 1C). As temperature decreases, the Arrhenius equation predicts $P_s$ to monotonically decrease. However, the mean fitted $P_s$ value at 10 °C was $0.541 \pm 0.030$ µm min$^{-1}$ but at 6 °C the mean $P_s$ value was $0.710 \pm 0.041$ µm min$^{-1}$. A possible explanation may be due to differences between oocytes and between treatments, as the treatments were done on separate days and with different females (three females per day). However, an alternative explanation is that the log of permeability parameter is nonlinear with respect to inverse temperature (contrary to the Arrhenius equation). Lower temperatures induce membrane phase transition that may result in a change of membrane permeability (*Quinn, 1985*; *Nedvěd, Lavy & Verhoef, 1998*; *Bayley et al., 2018*), and thus possibly explain the increase in relative permeability of $Me_2SO$ with respect to water. Future work may investigate novel nonlinear permeability models and experimentally validate them with a larger array of temperatures.

## Osmotic damage

The $J_{osm}(t;C_{osm},\beta)$ model fits osmotic damage throughout time with high accuracy (adjR$^2$>0.96; Figs. 2A–2C). Hypotonic solutions were less damaging than hypertonic solutions for the same volumetric deviance (swelling instead of shrinking). The high fitted powers of hypertonic solutions indicate sensitivity to shrinkage (Table 2). Furthermore, hypertonic exposure damage was influenced by whether the seawater included additional NaCl or sucrose. These results are in agreement with Davidson and colleagues (*Davidson et al., 2015*), who found that hypertonic sucrose solutions were more osmotically damaging for bovine endothelial cells than NaCl solutions in general. Our time-dependent osmotic damage model suggests that both solution type and time spent under osmotic challenge are important factors in osmotic survival. It is also very interesting to note that long exposure to hypertonic solutions cause artificial parthenogenesis in sea urchin species (*Loeb, 1900*). For *P. lividus*, artificial parthenogenesis causes a separation of the fertilization envelope (physically inhibiting fertilization), and increase in developmental abnormalities (*Von Ledebur-Villiger, 1972*). This phenomenon of osmotically induced artificial parthenogenesis may be the cause of the observed high sensitivity to hypertonic challenge.

Static osmotic tolerance limits may be a useful metric and are reported for many cell types, many of these reports do not investigate the time-dependent nature of osmotic damage (*Kashuba Benson, Benson & Critser, 2008*; *Blässe et al., 2012*). However, time-dependent osmotic damage has been reported in several cell types (*Zawlodzka & Takamatsu, 2005*; *Liu et al., 2009*; *Wang et al., 2011*; *Traversari et al., 2022*). If the mechanism of osmotic damage is due to ultrastructural changes in the lipid membrane and cytoskeletal matrix, then time-dependent osmotic damage may be a phenomenon common across all cells. Indeed, using an osmotic tolerance limit with an equilibration time of 5 min may provide a poor approximation of cell survival over the course of an hour. Conversely, osmotic tolerance limits taken at 10 min may provide little predictive power in terms of cell survival on shorter time scales.

Future work investigating the mechanisms of osmotic damage may include the role of membrane and cytoskeletal regulation. If damage is dependent on membrane and/or cytoskeletal changes throughout time, then we expect a decrease in temperature to reduce time-dependent damage, possibly following an Arrhenius relation. It is possible that both the fitted constant, $C_{osm}$ and the fitted power, $\beta$, are related to temperature *via* the Arrhenius relation, as we found with cytotoxicity damage. Indeed, *Wang et al. (2011)* found accumulated osmotic damage was temperature specific while *Williams & Takahashi (1982)* found membrane lysis of sea urchin eggs was mitigated at lower temperatures during hypertonic challenge. Supplements that prevent membrane and cytoskeletal regulation may reduce time-dependent osmotic damage (*Asadi, Najafi & Benson, 2022*). However, there may be a trade-off between time-dependent osmotic damage and sensitivity to lysis during osmotic challenge, or loss of cell functionality with supplements that depolarizes F-actin, such as cytochalasin-D or cytochalasin-B (*Fujihira, Kishida & Fukui, 2004*; *Takamatsu et al., 2005*; *Wang et al., 2016*). Furthermore, some somatic cells such as MDCK cells have zero survival to hypotonic shock when treated with cytochalasin-D but have an increased survival when treated with phalloidin (which stabilizes F-actin) (*Clegg, 1992*).

## Temperature damage

For short time periods (tens of minutes) chill injury is best predicted with respect to environmental temperature and not time of exposure as described by the $J_{\text{temp}}^2(t,T;C_{\text{temp}},T_{\text{phys}})$ model (Eq. (11)). The $J_{\text{temp}}^1(t,T;C_{\text{temp}},T_{\text{phys}})$ model (Eq. (10)) is built on the hypothesis that chill injury is both time and temperature dependent, however, we refuted this hypothesis on the time scale of tens of minutes since survival is not correlated to time at any temperature ($p>0.05$). Notably, the $p$-value of room temperature for the spearman ranked test is 0.071, we argue this is a statistical coincidence since we do not expect oocyte aging to be influenced within an hour and a half in seawater since room temperature is not expected to be a damaging temperature. Indeed, spawning events may last several hours at temperatures around 20 °C with no effect on development rate (*Boudouresque & Verlaque, 2013*).

Long-term (several hours to days) temperature damage is known to be time-dependent for somatic cells (*Nedvěd, Lavy & Verhoef, 1998*; *Bayley et al., 2018*) and sea urchin (*Hemicentrotus pulcherrimus*) oocytes (*Ohyama & Asahina, 1972*). While the mechanism of damage is poorly understood, it is known that a change in calcium permeability is the cause of chill injury for some somatic cells (*Bayley et al., 2018*). At low temperatures, lipid membranes may undergo a phase transition (*Quinn, 1985*). This phase transition is accompanied by a change in permeability to ions such as calcium (present in seawater). Sea urchin oocytes are particularly sensitive to calcium permeability changes as a small local flux of calcium may cause a cascading effect of opening voltage-sensitive channels on the membrane and calcium induced calcium release from the endoplasmic reticulum, resulting in depolarization of the cell and a calcium wave (*Shen & Buck, 1993*). This calcium wave is a ubiquitous signal in oocytes and follows when the sperm head unites with the oocyte membrane (*Whitaker, 2006*). Subsequently, the fertilization envelope lifts from the egg surface and acts as a barrier to avoid polyspermy (*Gardner & Evans, 2006*; *Cheeseman et al., 2016*). However, if this calcium wave is caused by chilling, then oocytes may lose the ability to be properly fertilized and develop to a normal 4-arm pluteus stage larvae, a phenomenon termed artificial parthenogenesis as mentioned above (*Loeb, 1900*; *Von Ledebur-Villiger, 1972*; *Stricker, 1999*).

## Cytotoxicity

Cytotoxicity is best described by considering CPA osmolality at the cell boundary, as captured by the $J_{\text{tox}}^3(t,T;C_0,E_{\text{tox}},\alpha_0,E_\alpha)$ model (Eq. (8)). This model outperforms both the $J_{\text{tox}}^1(t,T;C_0,E_{\text{tox}},\alpha)$ model (Eq. (6)), as well as the $J_{\text{tox}}^2(t,T;C_0,E_{\text{tox}},\alpha_0,E_\alpha)$ model (Eq. (7)) which has the same number of parameters. The theory driving the $J_{\text{tox}}^1$ and $J_{\text{tox}}^2$ models is that cytotoxicity is a metabolically driven phenomenon wherein CPAs interact with intracellular enzymes, proteins and organelles and hampers critical physiological processes. The $J_{\text{tox}}^3$ model, on the other hand, only considers the rate of changes on the cell membrane, such that lysis, pore formation, and changes in ion permeabilities are correlated with extracellular CPA at the cell boundary (*He et al., 2012*; *Fernández & Reigada, 2014*). Indeed, if the mechanism of chill injury for oocytes is a change in permeability to calcium and a subsequently artificial parthenogenesis with an accompanied loss in development

to normal four-arm pluteus stage larvae (*Loeb, 1900*; *Von Ledebur-Villiger, 1972*; *Stricker, 1999*).

Interestingly, the power function (*i.e.*, $\exp[\alpha_0 - E_\alpha/RT]$) for the $J_{tox}^3$ model is negatively correlated with temperature (such that lower temperatures have lower power constants) whereas the proportionality function (*i.e.*, $\exp[C_0 - E_{tox}/RT]$) is positively correlated with temperature (lower temperatures have higher proportionality constants). While the decrease in power indicates a reduction in sensitivity, the increase in proportionality indicates an overall increase in damage. For sea urchin oocytes, $Me_2SO$ is overall more damaging at lower temperatures than at physiological temperatures even after accounting for chill injury (Figs. 2D–2F). This is contrary to expected if the mechanism of CPA damage is metabolically driven (*Benson, Kearsley & Higgins, 2012*; *Davidson et al., 2015*). It is possible that at lower temperatures membrane pores created by $Me_2SO$ last longer, while repair to the sites in the lipid membrane is reduced due to phase transition and low temperatures. This mechanism may also explain the observed increase in $P_s$ at 6 °C (Fig. 1). Further, if these pores enlarge and stay enlarged for longer, then the likelihood of a calcium wave initiation may increase, resulting in higher rates of artificial parthenogenesis (and thus the lowered survival rates observed). Further research may investigate these and other possible mechanisms.

## Optimization modelling

Successful cryopreservation of sea urchin oocytes has yet to be reported. Our numerical optimization, covering the spectrum of minimal time, minimal volume change, and minimal cytotoxicity, shows that there is no current method of loading oocytes with enough CPA to obtain successful vitrification by either multi-step or continuous loading approaches (*Benson, 2015*; *Davidson et al., 2015*; *Benson et al., 2018*). High sensitivity to CPA and osmotic shock make the sea urchin oocyte very difficult to cryopreserve. Similar difficulties were found when developing cryopreservation protocols for oocytes of other marine invertebrates (*Paredes, 2016*; *Diwan et al., 2020*). Possible alternative methods of cryopreservation include liquidus tracking (*Kay et al., 2015*) and optimized slow cooling (*Mazur, Leibo & Chu, 1972*; *Bahari et al., 2018*), but it is unclear if urchin oocytes cryopreserved with these methods will be recoverable.

A key outcome of the damage models is identifying membrane sensitivity to temperature, dimethyl sulfoxide, and osmomechanical stress. This raises the hypothesis that artificial parthenogenesis is a key contributor to sea urchin oocyte sensitivity during cryopreservation and possibly other species as well (*Loeb, 1900*; *Mattioli et al., 2003*; *Larman, Sheehan & Gardner, 2006*; *Wang et al., 2017*; *Sanaei et al., 2018*). Future work may investigate artificial parthenogenesis in the context of cryo-related damages (osmomechanical, chill, and cyotoxic) and methodologies of blocking artificial parthenogenesis. Bovine oocyte cryopreservation is improved with the use of calcium chelators and ion channel blockers (*Wang et al., 2017*; *Sanaei et al., 2018*). These additions reduce the likelihood of a calcium wave initiating and artificial parthenogenesis (*Gardner & Evans, 2006*; *Whitaker, 2006*; *Wozniak & Carlson, 2020*). Blocking loss of functionality may be a key step in successful cryopreservation of sea urchin oocytes.

The determination of the appropriate equations for $J_{tot}$ may be used for other cells and enable a more accurate and robust search for cryopreservation protocols. These models provide insight into damages and help pinpoint mechanisms preventing cryopreservation. For instance, some cell types may not be chill sensitive, while others may have cytotoxicity better described by intracellular molality. Furthermore, we argue that the utility of constant osmotic tolerance limits must be tested in light of a trade-off between time-dependent osmotic damage and cytotoxicity during loading and unloading of CPA.

## CONCLUSIONS

We provide a novel model that rationally combines volumetric effects, solution effects, and temperature effects to predict total population survival throughout time. Existing approaches for modelling osmotic damage, cytotoxicity, and chill injury have been found to be inadequate or non-existent in the case of sea urchin oocytes. Our novel models provide a robust approximation of cell survival across time, temperature, and osmotic conditions, and highlight important mechanisms of damage that are often left uninvestigated. Future work may pinpoint these mechanisms of damage, providing insight into cellular stresses, adaptations, and responses.

## ACKNOWLEDGEMENTS

We graciously thank the excellent review and editing support provided by Robyn Shuttleworth. We graciously thank the aid provided by Sara Campos in performing osmotic damage experiments.

### Funding
This work was supported by NSERC: RGPIN-2017-06346, NSERC PGS-D, Gabriel Dumont Institute, and the Dept. Biology University of Saskatchewan. The funders had no role in study design, data collection and analysis, decision to publish, or preparation of the manuscript.

### Grant Disclosures
The following grant information was disclosed by the authors:
NSERC: RGPIN-2017-06346.
NSERC PGS-D, Gabriel Dumont Institute, and the Dept. Biology University of Saskatchewan.

### Competing Interests
The authors declare there are no competing interests.

### Author Contributions
- Dominic J. Olver conceived and designed the experiments, performed the experiments, analyzed the data, prepared figures and/or tables, authored or reviewed drafts of the article, and approved the final draft.

- Pablo Heres performed the experiments, authored or reviewed drafts of the article, and approved the final draft.
- Estefania Paredes conceived and designed the experiments, performed the experiments, authored or reviewed drafts of the article, and approved the final draft.
- James D. Benson conceived and designed the experiments, analyzed the data, prepared figures and/or tables, authored or reviewed drafts of the article, and approved the final draft.

## Data Availability

The Mathematica code and raw data are available in the Supplementary Files.

## Supplemental Information

Supplemental information for this article can be found online at http://dx.doi.org/10.7717/peerj.15539#supplemental-information.

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
