# Peer review of "Rational synthesis of total damage during cryoprotectant equilibration: modelling and experimental validation of osmomechanical, temperature, and cytotoxic damage in sea urchin (Paracentrotus lividus) oocytes"

_PeerJ, doi:10.7717/peerj.15539_

## Round 0.1 · original submission · Minor Revisions

All reviewers generally praised the manuscript. Nevertheless, there are several issues that need to be addressed by the authors before the final decision could be made. Please check and follow the recommendations suggested by the reviewers.

Reviewer 1 ·

Basic reporting

This manuscript is written in clear and professional English, and as such, the rationale for the project is clear. There are a handful of small spelling errors, the most common of which is missing use of hyphens (which makes some of the sentences less clear than they could be). Aside from this, the text is professional and the content has clearly been well researched, as demonstrated by the appropriate use of a wider range of course in the text. There is clear, technical details demonstrated throughout the introduction which demonstrates a clear context to the work.
The article text is well structured, in-keeping with the stylistic requirements of PeeJ. I would however draw attention to the reference style, which does not yet meet the style requirements of the journal. Many analysis files have been provided which correspond to the analyses that have been conducted. On inspecting these, they do not always have headings and while the majority of columns are self-explanatory, headings would help to clarify the data.

Experimental design

The content of the manuscript fits well within the remit of PeerJ. The rationale for the study is clearly stated, with well-considered background. The challenges that are currently being faced in cryopreservation are identified well and it is clear how this manuscript fits a gap within this field.
Methods are explained clearly and in a level of detail that encourages repeatability with regards to measurements, animal dissection and tissue sample protocol. Methods are lengthy, but this is a necessity in order to explain the main study components. Consideration to animal ethics has been provided in the work. It would be worth stating how assumptions for test choice (i.e. t test) were met.

Validity of the findings

This study covers three different experimental areas and has generated a large sample of data. In the methods, there is reference to some samples being discarded if methods were not followed in full, so a breakdown of the number of samples for each treatment should be provided. The study findings are clearly provided, and are well interpreted in the discussion and abstract. It would be nice to see consideration of replication and future studies in greater detail in the discussion, as the authors are in a good position to identify future avenues, given that sea urchin cryopreservation is currently unachievable.
Conclusions are clearly stated, providing a well considered overview of the study findings.

Additional comments

Overall, this is a well-considered, technical experimental study. The manuscript is written to a standard that is largely repeatable, though it would be nice to see further exploration of the meaning and future directions for the study. I would recommend revision of references to be in-keeping with the in-house style, checking of the points relating to test choice, and a brief proof read. These are relatively small points overall. Thank you for submitting this insightful study!

Annotated reviews are not available for download in order to protect the identity of reviewers who chose to remain anonymous.

Reviewer 2 ·

Basic reporting

This manuscript contains a clear professional description of a body of work to investigate the complex questions surrounding how cells survive cryopreservation by manipulations to load essential cryoprotectants. The authors are congratulated for their careful work to model the multiple damage factors for CPA manipulations which include temperature, CPA toxicity, osmotic damage and time. The cited literature is appropriate and the authors have taken care to explain how the modelling was carried out, and the limitations of the approaches. The figures and tables are very good. This is one of the first reports to address these factors on an interaction basis, and makes a significant contribution to the understanding of fundamental cryobiology.

Experimental design

The work is appropriate for the journal; the research questions are well defined and rigorous reporting of both data and the models applied are provided. Methods are very clearly described and where necessary the limitations are explained.

Validity of the findings

This study is unique in that it combines findings of different damage models known in cryobiology into an overarching total injury model. This is a significant contribution to the field and to the wider topic of kinetics of interacting cell injury mechanisms in other areas of toxicology. The conclusions are well laid out and relate very clearly to the original research questions.

The one and only comment I have is that the title is not quite correct. Whilst the work is focused on cryopreservation, at no point in this study was cryopreservation carried out. This was a study to investigate multiple factors of injury in the exposure of cryoprotectants towards eventual understanding of cryopreservation. The authors should consider a very minor change to the title and Abstract.
I have no suggestions for any improvements in the manuscript and have nothing else to say.

Additional comments

This is a very interesting study in a very complex area of fundamental cryobiology.

Reviewer 3 ·

Basic reporting

A very clear and details write up. A bit of lacking on the introduction part; research background in related to temperature exposure.
Concise write up as a whole manuscripts.

Experimental design

No comment.
Methods are details and informative

Validity of the findings

The data provided are robust and have element of novelty

Additional comments

This manuscript is ready to be published.
Some of the comments noted in pdf file

Annotated reviews are not available for download in order to protect the identity of reviewers who chose to remain anonymous.

---

## Round 0.2 · accepted · Accept

The authors have addressed all of the reviewers' comments as stated by the reviewers (2nd revision). The manuscript is now ready for publication, pending the final typesetting set up.

Reviewer 1 ·

Basic reporting

The revised manuscript demonstrates that the authors have addressed initial readability points, including use of hyphens and wording. The manuscript is clearly informed by the wider literature as demonstrated at the initial review, and is well formatted to the requirements of PeerJ.

Experimental design

No changes have been made to experimental design since the last review of the paper, but aspects of the methods have been explained more clearly. The research question is well defined and the tables used in the work are now much clearer with their headings added. The explanation regarding the choice of statistical test is not much clearer.

Validity of the findings

The question is clearly stated, methods are clearly explained and as such the work is described at a level of detail that would permit repeat studies. Conclusions are clearly stated but not overstated.

Reviewer 2 ·

Basic reporting

The authors have fulfilled the revision requirements. I have no further comments.

Experimental design

The authors have fulfilled the revision requirements. I have no further comments.

Validity of the findings

The authors have fulfilled the revision requirements. I have no further comments.

Additional comments

The authors have fulfilled the revision requirements. I have no further comments.